# Native European crayfish *Astacus astacus* competitive in staged confrontation with the invasive crayfish *Faxonius limosus* and *Procambarus acutus*

Ivo Roessink[1]*, Karina A. E. van der Zon[2,3], Sophie R. M. M. de Reus[2,4], Edwin T. H. M. Peeters[2]

1 Environmental Risk Assessment, Wageningen Environmental Research, Wageningen, The Netherlands, 2 Aquatic Ecology and Water Quality Management, Wageningen University and Research, Wageningen, The Netherlands, 3 Department of Zoology, University of Otago, Dunedin, New Zealand, 4 Advisory Group Ecology, Royal HaskoningDHV, Amersfoort, The Netherlands

* ivo.roessink@wur.nl

**Data Availability Statement:** All relevant data are within the manuscript and its Supporting Information files.

## Abstract

The European native, noble crayfish (*Astacus astacus)* has suffered from a serious and long term population decline due to habitat destruction, water pollution and the impact of the invasive North American crayfish that are carriers of the crayfish plague (*Aphanomyces astaci*). The latter being the major factor currently confining noble crayfish to uninvaded (parts of) waterbodies. However, recently wild populations of apparently healthy noble crayfish carrying the crayfish plague have been found. As crayfish are known for their inter- and intraspecific agonistic behaviour which may be key for their competitive success, this raised the interesting question what would happen if the crayfish plague would not be a dominant factor anymore in the interaction between native and invasive species. Since the outcome of those encounters is still unclear, this study explores whether the noble crayfish can stand its ground towards invasive species in such agonistic interactions. Furthermore, the ability of the noble crayfish and invasive crayfish to acquire shelter through agonistic interaction is also assessed. Through pairwise staged interactions, agonistic behaviour and shelter competition between the native *A. astacus* and the invasive *Faxonius limosus* and *Procambarus acutus* were examined. The results showed that *A. astacus* triumphs over *F. limosus* and *P. acutus* in agonistic encounters and in competition for shelter. In turn, *P. acutus* dominates *F. limosus* in staged encounters and shelter. In possible future situations were crayfish plague does no longer eradicate noble crayfish populations, our results show that the native noble crayfish might still have a promising future when confronted with invasive species.

## Introduction

Invasive species are one of the most dangerous threats to biodiversity worldwide [1], especially in freshwater ecosystems. Among the most successful freshwater ecosystem invaders are

**Funding:** The author(s) received no specific funding for this work.

**Competing interests:** The authors have declared that no competing interests exist.

crayfish [2] and through direct competition and as vectors of disease they can displace native crayfish, leading to population declines and local extinctions [3]. From approximately 1860 onwards, outbreaks of the crayfish plague have had a devastating effect on native crayfish populations in Europe [4]. The crayfish plague is caused by the oomycete *Aphanomyces astaci* (Schikora, 1903) that entered Europe through the introduction of North American crayfish [5–7]. While North American crayfish can live in a balanced host-parasite relationship with this *A. astaci*, the parasite is acutely pathogenic to European species [4, 8].

Europe's most common and economically most valuable indigenous crayfish species, the noble crayfish (*Astacus astacus* (Linnaeus, 1758)) has suffered from a serious and long term population decline due to habitat destruction and water pollution in combination with the crayfish plague [9]. The recent observations of apparently healthy noble crayfish carrying *A. astaci* [10–12] are therefore very remarkable. Several explanations for these findings have been suggested, including the possibilities that the As-genotype of the pathogen is becoming less virulent [10, 12], that less virulent strains of the pathogen are becoming more common [10] and that the resistance of *A. astacus* against the pathogen, or at least against the As-genotype of the pathogen, is increasing [11].

Whatever the reason, these findings trigger the interesting question: what would happen in the event that the crayfish plague presence or absence would not be the determining factor for native crayfish occurrence anymore? Would in that case, agonistic interactions and competition for shelter become more important determinants of competitiveness of *A. astacus* against invasive species? Agonistic, fighting related, behaviours are stereotyped [13] and largely conserved among crayfish species [14]. However, levels of aggression and outcomes of interspecific agonistic interactions differ among species [15], which can influence the outcome of interspecific competition [2]. This is partly because agonistic interactions influence a species ability to hold key resources such as shelters [16], which are critical for crayfish survival, providing protection against predators and refuge during vulnerable life stages and times of environmental stress [17].

At present, there are at least 10 non-indigenous crayfish species that *A. astacus* could encounter in Europe [3, 18]. One of the most successful invasive crayfish species in Europe is the North American signal crayfish *Pacifastacus leniusculus* (Dana, 1991; [16]). Although at the time of experimenting the species was difficult to obtain in the Netherlands, previous research by Söderbäck [16] has shown already that it dominates over *A. astacus* in agonistic interaction. As a result, there was no need to include it in our experiments. In contrast, agonistic interactions between *A. astacus* and the other 9 non-indigenous crayfish species have not been studied yet, nor has competition for shelter between *A. astacus* and non-indigenous crayfish. In order to fill part of this knowledge-gap we tested agonistic encounters and shelter occupancy using the native noble crayfish (*A. astacus*) and the invasive spiny-cheek crayfish (*Faxonius limosus* (Rafinesque, 1817), formerly *Orconectes limosus*) and white river crayfish (*Procambarus acutus* (Girard, 1852)), two invasive crayfish species readily available in The Netherlands [18].

*Faxonius limosus* was the first non-indigenous crayfish species to be introduced in Europe in 1890 [3] and has since established itself in 22 European countries [18]. *Faxonius limosus* is a well-studied vector of the crayfish plague [19]. A recent addition to the European crayfish fauna is the North American white river crayfish (*Procambarus acutus* (Girard, 1852)). In 2005, the first established population of *P. acutus* was recorded in the Netherlands and in 2012 an established population was also documented in the United Kingdom [18].

In the laboratory, paired species experiments were performed to compare outcomes of interspecific agonistic encounters [16–18, 20–25] and the ability of competing species to obtain shelter [16, 19, 20, 22]. We hypothesise that *F. limosus* will be less successful than the other two

species in agonistic interactions and competition for shelter because *F. limosus* is known to be low in aggression [26, 27]. Aggressive behaviour is common in both *A. astacus* [16] and *P. acutus* [20] but there is no literature available on dominance in agonistic behaviour and competition for shelter between the two species. Therefore, we hypothesise that both species have equal changes in aggressive encounters and competition for shelter.

## Materials and methods

*Astacus astacus* were obtained from a breeder in Germany (Harald Groβ, location Bad Münstereifel-Schönau; batch FZ2015/10) while *F. limosus* and *P. acutus* were wild-caught by a Dutch commercial fisherman (Blokland B.V., location Hardinxveld-Giessendam; batch 2015.0609.1&2015.1209.1). Crayfish were kept under controlled conditions (water temperature 21 ± 1°C, 8:16 h L:D regime, fluorescent light, pH 7.8–8.1) and were fed 2 to 4 Trouvit™ fish food pellets twice a week. Each individual stayed in a (10 L x 38 W x 38 H cm) section of an aquarium that was separated by perforated plastic dividers. The health status of the crayfish was daily checked. All crayfish used in the experiment were in intermoult stage, had fully intact appendages and showed no abnormal behaviour.

### General set-up experiment

In September and October 2015 experiments were performed in (50 L x 30 W x 30 H cm) glass aquaria with 1 cm of gravel and 15 cm of water that stood in a water bath. Interaction pairs of crayfish of the same gender and similar body size were selected in three species combinations:

1. *A. astacus* and *F. limosus*, 5 male and 5 female pairs

2. *A. astacus* and *P. acutus*, 9 male and 6 female pairs

3. *F. limosus* and *P. acutus*, 12 male and 6 female pairs

Differences in carapace length between the interaction pairs (Fig 1 and S1 Table) were mostly small except for pairs of *A. astacus* and *P. acutus*. Five out of nine *A. astacus* and *P. acutus* males had similar carapace lengths (< 3 mm difference) but carapaces of the *A. astacus* females were always > 9 mm longer than those of their *P. acutus* opponents. The difference in carapace length between female *F. limosus* and their *P. acutus* opponents was significant according to the two sided Wilcoxon signed rank test but always < 1.2 mm (Fig 1).

To minimize the chance of crayfish plague infection, experiments with *F. limosus* and *P. acutus* were performed first and thereafter those with *A. astacus. Astacus astacus* were kept separate from the other crayfish in a different isolated tank with a separate water circulation system. Different sets of equipment were used for *A. astacus* and for the other species and experimental aquaria were disinfected with Virkon S™ between trials. Also to avoid infection, individuals of *A. astacus* were only used once and were not returned to the holding tanks after the trial. Some *F. limosus* and *P. acutus* individuals, however, were used in a second trial that took place two or three weeks after the first. This period is long enough to not affect the behaviour of the crayfish in a second trial [25].

The interactions were examined in a two-staged experiment. In the first stage, agonistic behaviour between the two interacting individuals was observed for one hour. Over the following night, shelter occupancy was examined with the same two individuals in the same aquarium.

Before an experiment started, a plastic divider was placed in the middle of the aquarium and the two individuals of each pair were placed on opposite sides of the divider to acclimatise for 10 minutes. The divider was removed after the acclimatisation period and agonistic

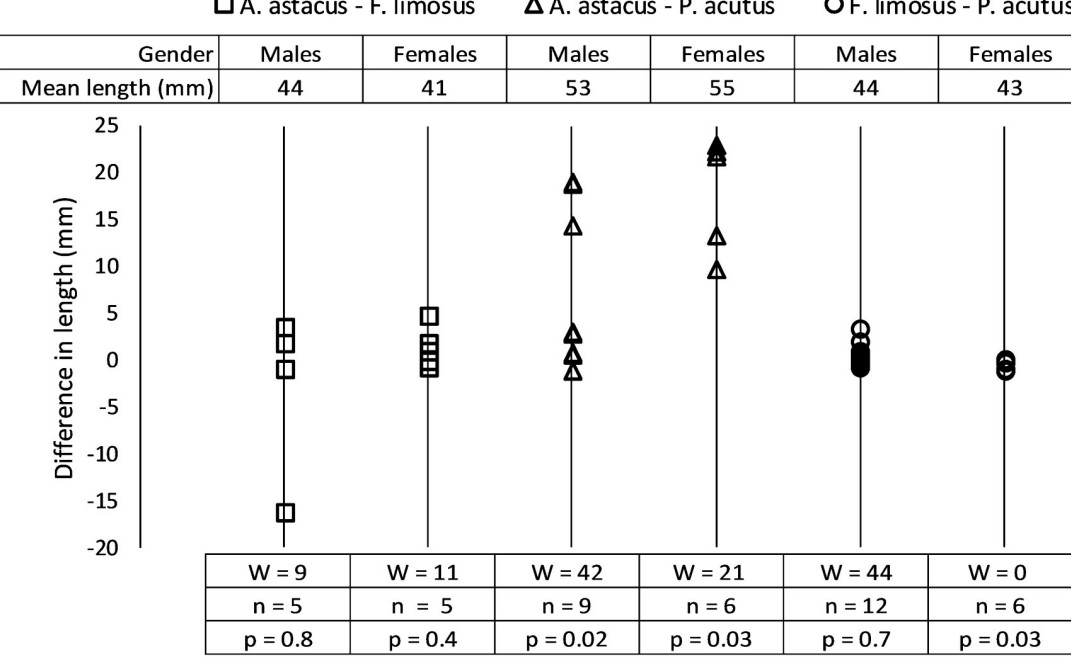

**Fig 1. The difference in carapace length between the two interacting crayfish.** The difference is positive when the specimen of the species named first in the legend is larger than the specimen of the species named second and negative when it is the other way around. Each data point indicates one interacting pair and the symbols are vertically aligned according to gender and species combination. Outcomes of two-sided Wilcoxon signed rank tests are indicated below the corresponding gender and species combination and the average carapace lengths of the trialled animals are indicated above the data points.

interactions between the two crayfish were observed by a single observer for 60 minutes. The following events were recorded:

1. *First attacks*: the number of fights started per individual with a fight defined as a crayfish using claws trying to touch or grab the opponent [28, 29].

2. *Wins*: number of times a crayfish stands its ground and forces the opponent to stop moving or retreat.

3. *Retreats*: number of times a crayfish moves away from the opponent for a distance equal to or larger than its body length [28]

After the 60 minute observation period, the crayfish were separated by replacing the divider between them in the middle of the aquarium. At around 5 pm, the divider was removed again and a (15 L x 10 W x 5 H cm) PVC tube was placed in the middle of the aquarium as a shelter. The next morning it was observed which of the two crayfish had occupied the shelter.

As a control, the shelter occupancy test was performed with single crayfish in the aquarium to ascertain that the crayfish desired to occupy the shelter [14]. This control was performed with 7 *F. limosus* males, 6 *F. limosus* females, 5 *P. acutus* males and 5 *P. acutus* females. Due to risk of contamination with the crayfish plague this control was not performed with *A. astacus*.

## Data analysis

Generalized Linear Model (GLM) analysis was performed to test whether gender was a significant variable explaining the number of agonistic interactions per pair. Gender, species identity and the interaction between gender and species were used as independent variables and an

intercept was calculated for each gender. Furthermore, a quasipoisson model with a logarithmic link function was chosen because the count data was overdispersed.

Each species' average number of first attacks, wins and retreats per trial was calculated for males and females separately based on the data in S1 Table. Because no obvious differences in behavioural patterns between the genders of the same species were observed, statistical analyses of interspecific outcomes were performed for combined male and female data. Two-tailed Wilcoxon signed rank tests were used to test for significant differences in the number of first attacks, wins and retreats between the interacting species. Pearson chi-square tests were performed to evaluate whether differences between expected and observed frequencies of shelter occupancy were significant. Expected frequencies were calculated based on the assumption that both species would occupy the shelter equally often. Statistical analyses were performed in R 3.6.3 using RStudio version 1.2.5042 [30].

## Results

### Agonistic interactions

In general, fewer agonistic interactions took place in female pairs than in male pairs (Fig 2) and patterns in agonistic behaviour between paired species did not seem to differ for male and female pairs. For the total number of agonistic interactions gender was significant (GLM $p \leq 0.002$, S3 Table). In male *A. astacus* and *P. acutus* pairs, there was a higher number of first attacks than in male *A. astacus* and *F. limosus* pairs, mostly due to a higher number of first attacks by *A. astacus* in the presence of *P. acutus* than in the presence of *F. limosus*. Female *A. astacus* also started more fights than *F. limosus* and *P. acutus* but there were not more first attacks by female *A. astacus* in presence of *P. acutus* than in presence of *F. limosus*.

For the *A. astacus* vs. *F. limosus* pairs, there were more first attacks (two-tailed Wilcoxon signed rank test, $W = 44$, $n = 9$, $p = 0.01$), more wins ($W = 28$, $n = 7$, $p = 0.02$) and less retreats ($W = 1.5$, $n = 9$, $p = 0.02$) by *A. astacus* than by *F. limosus*. In the *A. astacus* vs. *P. acutus* pairs, *A. astacus* performed more first attacks ($W = 108$, $n = 15$, $p = 0.007$), won more fights ($W = 91$, $n = 13$, $p = 0.002$) and retreated less ($W = 7$, $n = 13$, $p = 0.003$) than *P. acutus*. Lastly, in the *F. limosus* vs. *P. acutus* pairs there were more fist attacks ($W = 13.5$, $n = 16$, $p = 0.005$) and wins ($W = 21$, $n = 14$, $p = 0.05$) by *P. acutus* than by *F. limosus*, but the difference in retreats was not significant ($W = 123$, $n = 18$, $p = 0.1$).

### Shelter competition

In the single animal shelter occupancy controls *F. limosus* males and females were found in the shelter in all trials. Also *P. acutus* females occupied the shelter in all trials. Only the *P. acutus* males were found outside of the shelter in 4 of the 7 trials (S2 Table).

The shelter in the competition experiment was always occupied in the trials with *A. astacus* and *F. limosus* pairs but was sometimes found empty in the trials where *P. acutus* participated (Fig 3). *Astacus astacus* occupied the shelter more often (Chi-square test, $\chi^2 = 12.8$, $df = 1$, $p < 0.001$) than *F. limosus* and more often than *P. acutus* ($\chi^2 = 5$, $df = 1$, $p = 0.03$). There was no significant difference in shelter occupancy between *F. limosus* and *P. acutus* ($\chi^2 = 1.9$, $df = 1$, $p = 0.2$).

## Discussion

### Outcomes of the paired species experiments

The results show that of the three species *A. astacus* was the most aggressive and most successful in fight and shelter acquisition. *Procambarus acutus* was the next most aggressive and

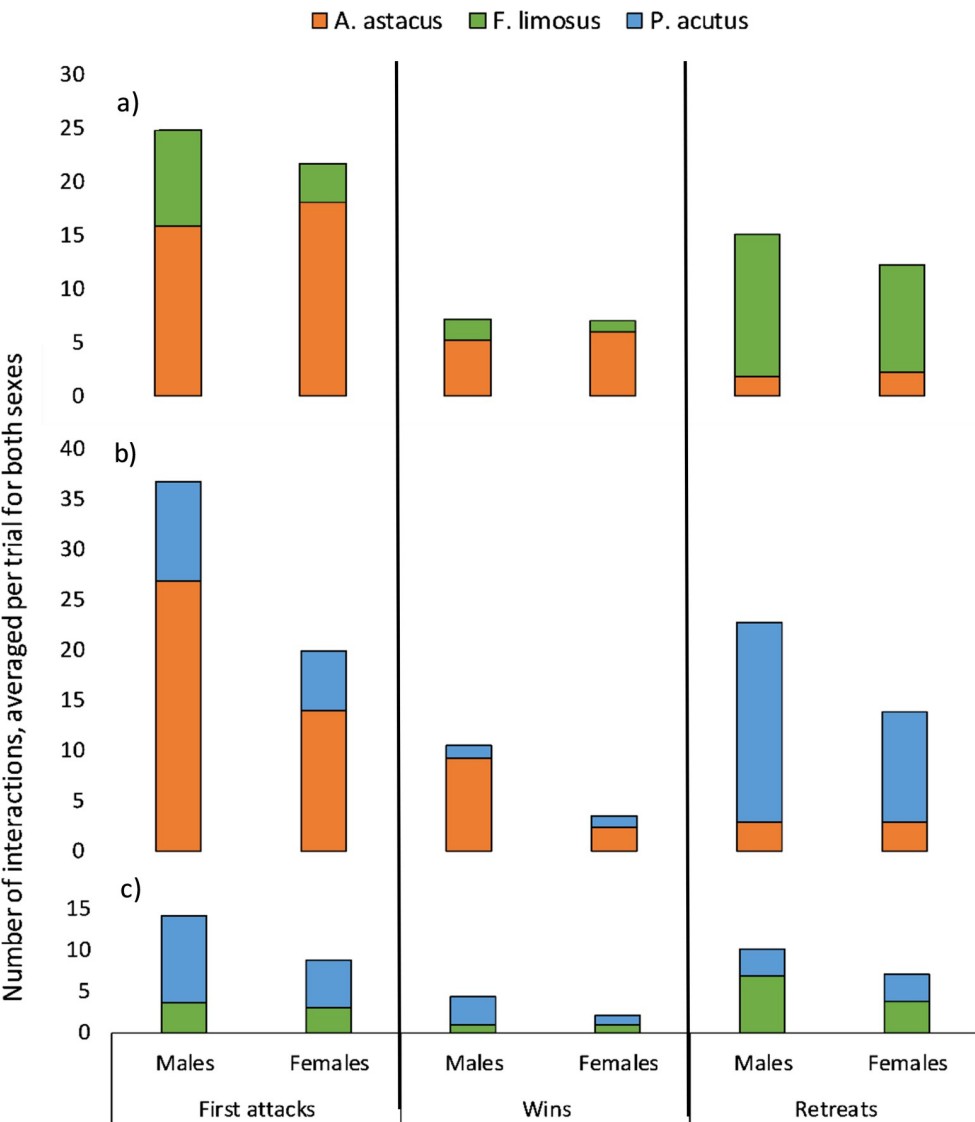

**Fig 2.** Average number of first attacks, wins and retreats per trial with a) *A. astacus* vs. *F. limosus*, b) *A. astacus* vs. *P. acutus* and c) *F. limosus* vs. *P. acutus* by males and females.

successful in agonistic interactions and *F. limosus* the least. There was a larger number of agonistic interactions between male pairs than between female pairs, which is in accordance with the general notion that male crayfish are more aggressive than female crayfish [31].

Body size has a major influence on crayfish dominance [32] and the difference in carapace length between *A. astacus* and their *P. acutus* opponents could have influenced the outcome of the experiment. However, when inspecting only the five *A. astacus* vs. *P. acutus* pairs that had a difference in carapace length that is <10% of the average carapace length of the two opponents, the same pattern arose as when considering all *A. astacus* vs. *P. acutus* pairs with *A. astacus* starting and winning more fights and retreating less than *P. acutus*. Interestingly, there was one *A. astacus* that was shorter than its opponent but still dominated the encounters.

The shelter occupancy control experiment showed that *F. limosus* males and females and *P. acutus* females preferred to be in the shelter and although the control was not performed with

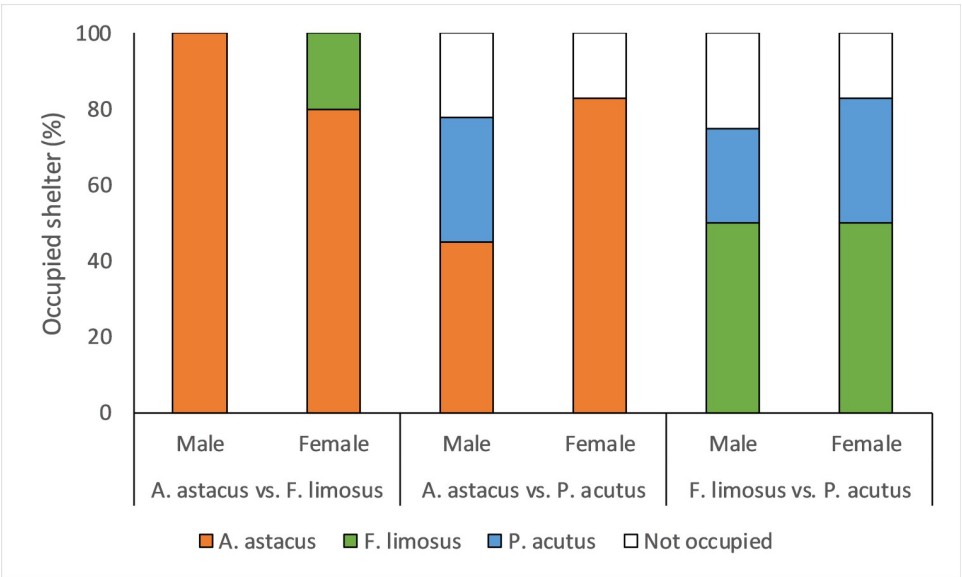

**Fig 3. Shelter occupancy by the interacting species in percentages for males and females.** Numbers of trialled male and female pairs of each species combination are indicated above the bars.

*A. astacus*, it is clear from the outcomes of the paired species shelter occupancy experiment that *A. astacus* males and females stayed in the shelter as well. It is interesting to notice that the shelter in the trials with *P. acutus* was sometimes not occupied by either of the competitors. This might be because the animals were still fighting over the shelter at the time the occupancy was assessed. Unfortunately no video recordings of the experiment were made, so an additional check couldn't be made. An alternative explanation for the unoccupied shelters could be that the dominant crayfish prevented the subordinate one from using it as was also observed by Gherardi and Daniels [20] in a shelter occupancy experiment where the dominant *Procambarus clarkii* did not use the shelter after evicting subordinate *P. acutus acutus* from it.

## Relevance for crayfish populations in Europe

Remarkable changes in the host-parasite relationship between *A. astaci* and the native European crayfish species *Astacus leptodactylus* [31], *Austropotamobius torrentium* [32] and *Austropotamobius pallipes* [33] have recently been reported. *A. astaci* used to have a devastating effect on these species, but lately populations of these three species carrying *A. astaci* as a subclinical infection and showing melanized spots have been found. It is not inconceivable that in the long-term similar changes may occur in the host-parasite relationship between *A. astacus* and *A. astaci*.

Outside of the laboratory, differences in body sizes between species probably do matter for the outcome of interspecific agonistic encounters because *A. astacus* males, for example, can grow to a length of 180 mm and females can reach 150 mm while *F. limosus*, on the other hand, has a maximum length of 61 mm. Furthermore, specimens of *P. acutus* rarely grow longer than 140 mm [34]. However, the maximum length that a species can achieve is not the sole determinant of the size structure of a crayfish population. Juveniles of *F. limosus* grow much faster than juvenile *A. astacus* [30] giving them a competitive advantage at young age. Such an advantage is even bigger for *P. acutus* because this species has an even higher growth rate than *F. limosus* [35]. How these differences in juvenile development affect agonistic interactions

between *A. astacus* and invasive crayfish remains unclear and can be studied in experiments in which juveniles are reared together [30].

Furthermore, other traits besides agonistic dominance could contribute to the competitive success of a crayfish species over another. For example, the higher growth rate, higher egg production per capita, the lower age and smaller size at which *P. leniusculus* reaches sexual maturity were all important for the displacement of *A. astacus* by the more aggressive *P. leniusculus* in Swedish [36] and Finnish [37] lakes. Bearing this in mind, it is highly recommended to include *P. clarkii* (Girard, 1852) in further research on interspecific agonistic interactions between *A. astacus* and invasive species because *P. clarkii* is known to be aggressive [26], and is widespread in Europe and increasing its numbers [3]. Van Kuijk et al. [35] compared traits of successful and unsuccessful invasive crayfish in the Netherlands and found that temperature tolerances, egg counts, and numbers of clutches per year contributed to invasion successes. However, these researchers stressed that there are different routes to success [35]. For example, although *F. limosus* scores low on agonistic behaviour it is a highly successful invader [3] due toits parthenogenetic reproduction [3], fast population growth [30], and indifference to land use change [38].

In conclusion, several traits and combinations of mechanisms may explain the success of invasive crayfish and interspecific agonistic interaction and competition for shelter are relevant. While invasive crayfish continue to threaten European waterways [3] and the crayfish plague still has disastrous effects on *A. astacus* [12], recent observations [31–33] indicate the possibility that as other native crayfish are doing, in time the noble crayfish might also coexist with *A. astaci*. As a result, this justifies research on behavioural interactions between the noble crayfish and sympatric invasive crayfish species. In the wild, a lot of different factors determine the success of crayfish populations [35]. Of course, it remains to be seen whether the investigated endpoints in this study, e.g., winning direct interactions or the competition for shelter, are indeed key drivers. Once crayfish plague is no longer a decisive factor, long-term research on populations of invasive and native crayfish kept under controlled outdoor conditions could provide explanations to those specific questions. Nevertheless, the present study has clearly shown that in absence of the crayfish plague, the noble crayfish can at least behaviourally resist aggressive advances of two invasive species, *F. limosus* and *P. acutus*.

## Supporting information

**S1 Table. Agonistic interactions.** Number of "First Attacks", "Wins" and "Retreats" observed for each trialed crayfish.
(PDF)

**S2 Table. Shelter occupancy.** Frequencies of shelter occupancy for male and female species pairs.
(PDF)

**S3 Table. Summary GLM agonistic interactions.** Coefficients, standard errors, t-statistics and p-values of three quasipoisson distributed log linked Generalized Linear Models (GLMs) without intercept based on 43 observations for the dependent variables "First Attacks", "Wins" and "Retreats".
(PDF)

## Acknowledgments

The authors thank Lars Bakermans and Marie-Claire Boerwinkel for their assistance with conducting the experiments and Marc Schallenberg for proofreading. Also, the feedback from 3 reviewers is highly appreciated.

## Author Contributions

**Conceptualization:** Ivo Roessink, Edwin T. H. M. Peeters.

**Formal analysis:** Karina A. E. van der Zon, Edwin T. H. M. Peeters.

**Investigation:** Ivo Roessink, Sophie R. M. M. de Reus.

**Writing – original draft:** Karina A. E. van der Zon.

**Writing – review & editing:** Ivo Roessink, Edwin T. H. M. Peeters.

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
