## [Decision Letter · Decision Letter 0]

22 Jul 2021

PONE-D-21-20033

Native European crayfish Astacus astacus competitive in staged confrontation with two invasive crayfish species

PLOS ONE

Dear Dr. Roessink,

Thank you for submitting your manuscript to PLOS ONE. After careful consideration, we feel that it has merit but does not fully meet PLOS ONE’s publication criteria as it currently stands. Therefore, we invite you to submit a revised version of the manuscript that addresses all the points raised during the review process. However, there are major concerns by the reviewers and you have to carefully address all the comments by all three reviewers.

We look forward to receiving your revised manuscript.

Kind regards,

Irene Söderhäll

Academic Editor

PLOS ONE

Journal Requirements:

2. We understand that you purchased the different species of crayfish from local fishermen for this study. In your Methods section, please provide additional regarding the source of this material. Please provide the geographic coordinates and names of the purchase locations (e.g., stores, markets), if available, as well as any further details about the purchased items (e.g., lot number, source origin, description of appearance) to ensure reproducibility of the analyses.

Reviewers' comments:

Reviewer's Responses to Questions

**Comments to the Author**

1. Is the manuscript technically sound, and do the data support the conclusions?

Reviewer #1: No

Reviewer #2: Partly

Reviewer #3: Yes

2. Has the statistical analysis been performed appropriately and rigorously? 

Reviewer #1: No

Reviewer #2: Yes

Reviewer #3: Yes

3. Have the authors made all data underlying the findings in their manuscript fully available?

Reviewer #1: Yes

Reviewer #2: Yes

Reviewer #3: Yes

4. Is the manuscript presented in an intelligible fashion and written in standard English?

Reviewer #1: No

Reviewer #2: Yes

Reviewer #3: Yes

5. Review Comments to the Author

Reviewer #1: This manuscript decaying behavior of different crayfish species is in ite present form too premature and does not deserve to be included in this journal even after a revision.

The reference to that A.astaci originates from N.America is not number 5. Please give the correct and first reference to this discovery. .Also reference number 4 is not correct for the first discovery of this phenomen.

That A.astacus has become more resistant to crayfish plague is not performed in a proper way how this occurs in this paper,i.e showing why they are more resistant and thus one should be very careful with such statements. There are and always will be individual variability in resistance to crayfish plague between different populations or individuals so unless anyone can demonstrate that a population is more resistant because of any immune factor or immune process such statements should be avoided.The most obvious flaw with this paper is that P.leniusculus is not included as a species against all other crayfish species. Since this species is aggressive and thus must be included to really say something about behaviour between different species.

The obtained results are of very limited interest since all experiments are done in “in vitro” and thus do not say anything about what might happen in the wild. Further no information about the health status of the crayfish is provided. Crayfish were bought from commercial fishermen and thus we do not know how these crayfish and their health status are. Now it is also known that the microbiome is varying quite a lot between different individuals and this also affects their resistance to for example bacterial challenge ( newly published information). Wild crayfish and crayfish which have been in tanks or ponds may therefore have very different flora and this may also affect their behaviour as it does in most animals!

Therefore this piece of work is not scientifically sound especially since P.leniusculus is not included and also because references to work done on plague and crayfish are not correct and seems to have been chosen randomly rather chosen which paper showed it first.

.

Reviewer #2: Although potentially interesting the authors make very far-reaching conclusions when considering the limited amount of data those are based on. It is suggested that the text is modified somewhat with respect to the preliminary nature of their respect to reflect this. Also, the discussion regarding changes of pathogen resistance among indigenous crayfish needs some clarifications.

In total a modest number of 10 Astacus astacus individuals were tested for aggressiveness towards about similar numbers of the other two species. Presumably they were related and of similar age. While it seems plausible that this species is more aggressive than the other two, testing other individuals at other life cycle stages with respect to moulting, egg-bearing, age etc could have resulted in other outcomes and being of relevance for how the species interact with each other. The authors strongly stated conclusion appears a bit to simplistic.

The authors are stating that native crayfish are becoming more resistant towards crayfish plague. Available data – as based on the papers cited in the ms and a few more – show the existence of A. astaci strains with a lowered virulence (reference 9) and a few well-documented cases of variations in resistance between different populations of indigenous crayfish populations (especially reference 29). There are so far no experimental demonstrations of a change towards a higher A. astaci resistance among native populations although such changes may well take place in the long run. However, at this stage the conventional epidemiological model that a pathogen with a dramatically shorter generation time than the host is quicker to adapt its virulence (i.e. to lower it and not go extinct) should be over-looked. Some of the papers the authors cite in their discussion in this context, e.g. ref 28, actually argue for that pathogen virulence changes are a major factor at play here. The existence of a melanisation reaction could, as the authors suggest, be interpreted as a sign of host resistance but the other side of the coin is that melanisation can be a consequence of a lower parasite virulence that enables the host to mount a defence.

Reviewer #3: The manuscript entitled Native European crayfish Astacus astacus competitive in staged confrontation with two invasive crayfish species by Roessink et al deals with the ability of competing crayfish species to obtain shelter by a specific experimental design, and focuses on whether the native European A. astacus can defy the invasive

North American species F. limosus and P. acutus in agonistic behavior and competition for shelter. The investigation is justified within the context of the finding of native A. astacus showing increased resistance to the crayfish plague and envisioning of future scenarios of coexistence of both native and alien species.

I consider this study of great interest and needed for conservation of native crayfish. Therefore, I believe that this study deserves publication in a journal such as Plos One, which has a broad audience and is open access.

The manuscript is generally well written and structured. I find, however, some major comments to be considered by the authors listed next:

1. The objective and hypothesis of the study need to be more clearly written and emphasized in the introduction.

2. Line 51. It is said a notorious vector of the crayfish plague…. Since all vectors are notorious, I would erase this adjective or search for another one: well-known or well-studied.

3. Line 58 Europe`s most common and mostly highly valued indigenous crayfish is the noble crayfish…. I would said economically most valuable.

4. There is no section for figure legends and instead there is a figure legend for Fig 1 inserted in the text in pag 5.

5. Line 61 to 63. Be aware that the pathogen is not becoming less virulent. This only appears to apply for one haplotype of the pathogeni (haplotype -A but no for others, e.g, B, D1, D2 and E)

6. There are only three figures in the article, and there is no need for tables to be in a supplementary materials.

7. The quality of figures is low and there is no section for figure legends

8. The discussion misses the fact that the two predominant invasive crayfish species are Procambarus clarkii and Pacifastacus leniusculus and that similar studies considering these species are needed. It is true that they are mentioned but I would suggest further discussed this fact.

9. Make sure the scientific names of species are in italics in all the text, figures and references.

10. Please write the full names of the species when a sentence starts with a species name.

11. A number of references can be improved. I suggest the following:

- Statements for Lines 46, a more detailed and specific reference are in reviews by:

Söderhäll, K., and Cerenius, L. (1999). The crayfish plague fungus: history and recent advances. Freshw. Crayfish 12, 11–35.

Cerenius, L., Andersson, M.G., and Söderhäll, K. (2009). ”Aphanomyces astaci and crustaceans,” in: Oomycete Genetics and Genomics: Diversity, Interactions and Research Tools, eds. K. Lamour and D Kamoun (New Jersey: John Wiley & Sons Inc.), 425-433.

Rezinciuc, S., Sandoval-Sierra, J.V., Oidtmann, B., and Diéguez-Uribeondo, J. (2016). “The biology of crayfish plague pathogen Aphanomyces astaci: Current answers to most frequent questions”, in: Freshwater Crayfish – A Global Overview, eds. T. Kawai, Z. Faulkes, and G. Scholtz (London, UK, Taylor and Francis Group, CRC Press), 182–204.

- Statement starting in line 47 and 49. Original citations are:

Unestam, T. (1969b). Resistance to the crayfish plague in some American, Japanese and European crayfishes. Rep. Inst. Freshw. Res., Drott. 49, 202-209. doi: n/a

Unestam, T. (1972). On the host range and origin of the crayfish plague fungus. Rep. Inst. Freshw. Res., Drott. 52, 192-198. doi: n/a

Unestam, T., and Weiss, D.W. (1970). Host parasite relationship between freshwater crayfish and crayfish disease fungus, Aphanomyces astaci. Responses to injection by a susceptible and a resistant species. J. Gen. Microbiol. 60(1), 77-90. doi: 10.1099/00221287-60-1-77

Or recent publication:

Martín‑Torrijos L, María Martínez‑Ríos, Gloria Casabella‑Herrero, Susan B.Adams, Colin R. Jackson, Javier Diéguez‑Uribeondo. 2021.Tracing the origin of the crayfish plague pathogen, Aphanomyces astaci, to the Southeastern United States, Scientific Reports. 11:9332 | https://doi.org/10.1038/s41598-021-88704-8.

- Statement line 52

Original citation for presence of A. astaci in orconectes and its role as vector is:

Vey, A., Söderhäll, K., and Ajaxon, A. (1983). Susceptibility of the Orconectes limosus Raff. to the crayfish plague, Aphanomyces astaci Schikora. Freshw. Crayfish 5, 284-291. doi. 10.5869/fc.1983.v5.284

lines 203-207. requieres to be rewritten.

6. PLOS authors have the option to publish the peer review history of their article (what does this mean?). If published, this will include your full peer review and any attached files.

Reviewer #1: No

Reviewer #2: No

Reviewer #3: No

---

## [Author Response · Author response to Decision Letter 0]

4 Sep 2021

Response to reviewers

Dear professor Söderhäll,

Thank you for inviting us to submit a revised version of our manuscript entitled, “Native European crayfish Astacus astacus competitive in staged confrontation with two invasive crayfish species” to PLOS ONE. We would like to thank you and the reviewers for reading our manuscript and providing us with feedback. We aim to address all the points raised by you and by the reviewers in a point-by-point manner and use a blue text colour (see document response to reviewers) for our responses in the text below. 

Editor suggestions:

• Please ensure that your manuscript meets PLOS ONE's style requirements.

o Response: We reread the PLOS ONE style requirements and changed the figure labels and figure legends as to meet the requirements. These changes are detailed below in response to reviewer #3.

• In your Methods section, please provide additional regarding the source of this material [the purchased crayfish]. Please provide the geographic coordinates and names of the purchase locations (e.g., stores, markets), if available, as well as any further details about the purchased items (e.g., lot number, source origin, description of appearance) to ensure reproducibility of the analyses.

o Response: The requested information has been added for as far this was available. For instance, the coordinates of the precise location were the fisherman had caught the crayfish were not recorded as this is not common practise in crayfish fisheries in The Netherlands. This has been solved by mentioning the closest village near the catching site.

Reviewers’ responses to questions:

1. Is the manuscript technically sound, and do the data support the conclusions?

o Response: Reviewer #1 answered “No” to this question and reviewer #2 “Partly”. We understand from the comments of these reviewers that we should be more careful in drawing our conclusions. Therefore, we made changes in the wording of our conclusions, as detailed in this letter, and thereby ensured that they are supported by our data. 

2. Has the statistical analysis been performed appropriately and rigorously?

o Response: Reviewer #1 answered “No” to this question. However, reviewer #1 did not comment on the statistical analyses in his/her comments, so we do not know how to improve the statistics accordingly. Furthermore, reviewers #2 and #3 answered “Yes” to this question and since we’re still convinced of the appropriateness of the applied statistics ourselves, we decided to not change anything in response to this question. 

3. Have the authors made all data underlying the findings in their manuscript fully available?

o Response: Yes

4. Is the manuscript presented in an intelligible fashion and written in standard English?

o Response: Like to question 2, reviewer #1 answered “No” to this question but did not explain how the presentation or language should be improved in his/her comments. The manuscript was proofread by a native speaker and reviewers #2 and #3 answered “Yes” to this question so we did not make changes in response to this question.

Reviewers’ comments:

Reviewer #1:

• This manuscript decaying behavior of different crayfish species is in its present form too premature and does not deserve to be included in this journal even after a revision.

o Response: We acknowledge that future research is needed, but we think that the outcomes of our study are interesting and have scientific merit. We revised our manuscript and changed the conclusions as detailed below and we think that our revised manuscript deserves publication. We feel strengthened by the comments of Reviewer #3 who wrote: “I consider this study of great interest and needed for conservation of native crayfish. Therefore, we believe that with this revision this study deserves publication in a journal such as PLOS ONE, which has a broad audience and is open access.” 

• The reference to that A.astaci originates from N.America is not number 5. Please give the correct and first reference to this discovery. .Also reference number 4 is not correct for the first discovery of this phenomen.

o Response: We thank the reviewer for his/her suggestion to change references 4 and 5. As detailed in our response to reviewer #3 we changed them to the more appropriate references that were kindly suggested by reviewer #3.

• That A.astacus has become more resistant to crayfish plague is not performed in a proper way how this occurs in this paper,i.e showing why they are more resistant and thus one should be very careful with such statements. There are and always will be individual variability in resistance to crayfish plague between different populations or individuals so unless anyone can demonstrate that a population is more resistant because of any immune factor or immune process such statements should be avoided.

o We would like to thank the reviewer for pointing out that we seem to claim that A. astacus has become more resistant to the crayfish plague. We now saw that some parts of our manuscript, for example lines 27-29 in our abstract, were not formulated carefully. Our paper is not about the resistance of A. astacus against the crayfish plague. We examined agonistic behaviour and shelter competition. However, the relevance of our study is dependent on whether or not A. astacus will suffer less from the crayfish plague in the future and we can see that our use of language can be confusing for the reader so we clarified our objective and hypothesis of the study, as described in response to reviewer #3, and took care to avoid to seem to be claiming that A. astacus becomes more resistant to the plague. 

o We changed lines 27-29 of our abstract: “Recently, an increased resistance of the noble crayfish towards the crayfish plague has been observed, which means that the species now survives the disease and the presence of invasive crayfish. Potentially this may lead to actual agonistic encounters between native and invasive species” into “Recently, wild populations of apparently healthy noble crayfish carrying the crayfish plague have been found. In case that at some point in the future the proximity of plague carrying invasive crayfish will result more often in noble crayfish populations carrying the plague as a latent infection instead of in eradication, potential agonistic encounters between the native and invasive species may become more important.” 

o Also we changed lines 65-7: “(8,9), supporting the hypotheses that the pathogen is becoming less virulent (8) and that the resistance of A. astacus against A. astaci is increasing (9). In the event that a balanced host-parasite relationship between the noble crayfish and A. astaci evolves, agonistic interactions and competition for shelter may become the most important determinants of competitiveness of A. astacus against invasive species.” into “(12,13,14). Several explanations for these findings have been suggested, including the possibilities that the As-genotype of the pathogen is becoming less virulent (12,14), that less virulent strains of the pathogen are becoming more common (12) and that the resistance of A. astacus against the pathogen, or at least against the As-genotype of the pathogen, is increasing (13). In the event that the host-parasite relationship between the noble crayfish and A. astaci becomes more balanced, agonistic interactions and competition for shelter may become more important determinants of competitiveness of A. astacus against invasive species.” 

o We cited the following paper (14): “Makkonen J, Jussila J, Kortet R, Vainikka A, Kokko H. Differing virulence of Aphanomyces astaci isolates and elevated resistance of noble crayfish Astacus astacus against crayfish plague. Dis Aquat Organ. 2012;102(2):129–36.” This study shows that there are differences in virulence between different isolates of the plague and that different populations of the noble crayfish have different resistances towards some As-genotype isolates of the crayfish plague. At the end of this publication the authors speculate on signs of increased resistance towards the As-genotype in some populations that genetic adaptation within the As-genotype could have occurred since it first entered Finland. 

o The above publication by Makkonen et al. (2012), like the other publications we cited about changes in the host-parasite relationship between European crayfish and the crayfish plague keep stressing the danger of the plague. Therefore we also added the following words to our paper: “and the crayfish plague still has disastrous effects on A. astacus (14)” (line 266) 

o Furthermore we changed lines 276-277 from the discussion “the increasing resistance of the native A. astacus towards the crayfish plague might be cause for cautious optimism. This study has shown that in absence of the crayfish plague, the noble crayfish can behaviourally resist aggressive advances of two invasive species, suggesting that if A. astacus indeed manages to establish a balanced relationship with A. astaci, the species could displace F. limosus, P. acutus, and perhaps other invasive crayfish from its native habitat” into “ the possibility that in time the noble crayfish might coexist with A. astaci opens new perspectives. The recent discoveries of wild noble crayfish populations carrying latent infections of the crayfish plague justifies research on behavioural interactions between the noble crayfish and sympatric invasive crayfish species. This study has shown that in absence of the crayfish plague, the noble crayfish can behaviourally resist aggressive advances of two invasive species, F. limosus and P. acutus.” 

• The most obvious flaw with this paper is that P.leniusculus is not included as a species against all other crayfish species. Since this species is aggressive and thus must be included to really say something about behaviour between different species.

o Response: We agree with reviewer #1, as well as with reviewer #2 that it would have been very interesting to include Pacifastacus leniusculus in our study, especially since the study by Söderbäck from 1991 (which we cite in our paper) shows that P. leniusculus can dominate A. astacus in agonistic interaction. However, our study was performed in The Netherlands and P. leniusculus is quite rare in The Netherlands. We changed our introduction and included the information about P. leniusculus. We added lines “At present, there are at least 10 non-indigenous crayfish species that A. astacus could encounter in Europe (3,18). Previous research by Söderbäck (16) has shown that the North American invasive signal crayfish (Pacifastacus leniusculus (Dana, 1852)) dominates over A. Astacus in agonistic interaction. In contrast, agonistic interactions between A. astacus and the other 9 non-indigenous crayfish species have not been studied yet, nor has competition for shelter between A. astacus and non-indigenous crayfish. In order to fill part of this knowledge-gap we tested agonistic encounters and shelter occupancy using the native noble crayfish (A. astacus) and the invasive spiny-cheek crayfish (Faxonius limosus (Rafinesque, 1817), formerly Orconectes limosus) and white river crayfish (Procambarus acutus (Girard, 1852)), two invasive crayfish species readily available in The Netherlands (18).” 

• The obtained results are of very limited interest since all experiments are done in “in vitro” and thus do not say anything about what might happen in the wild. 

o Reviewer #1 is right that our experiment was “in vitro” however such experiment are extremely useful in finding cause-effect relationships or mechanisms. Due to many confounding factors exploring those cause-effect relationships in the outside world is a hard task. Field observations are needed too, but in order to focus on agonistic interactions a lab experiment seemed useful to us and we think that the results of our lab experiment could contribute to the scientific understanding of phenomena that may take place in the wild. We agree with the reviewer that our results explain only a part of the interactions that can occur in the outside world since many other factors are at stake that might negatively or positively affect our findings. In our discussion (lines 228-250) we discussed many aspects of the outside world that differ from the lab situation

• Further no information about the health status of the crayfish is provided. Crayfish were bought from commercial fishermen and thus we do not know how these crayfish and their health status are. 

o We thank the reviewer for this remark. We added the following sentence to our materials and methods section: “The health status of the crayfish was daily checked. All crayfish used in the experiment were in intermoult stage, had fully intact appendages and showed no abnormal behaviour.” (lines 83-85)

• Now it is also known that the microbiome is varying quite a lot between different individuals and this also affects their resistance to for example bacterial challenge ( newly published information). Wild crayfish and crayfish which have been in tanks or ponds may therefore have very different flora and this may also affect their behaviour as it does in most animals!

o We found a preprint of as study by Hernández-Pérez et al. (2021) which shows variation in the microbiomes of individuals of P. leniusculus that were obtained from the same lake. Although we do not know of any study which proves that differences in microbiome can affect the behaviour of crayfish, this hypothesis may be true. However, especially since we are not aware of research studying the effect of the microbiome on the agonistic behaviour of crayfish, we could not link this information into our manuscript. It was not feasible for us to account for the different microbiomes of the crayfish in our study. Furthermore, for the reintroduction and stocking of noble crayfish specimen from breeders are used thus justifying the combination of bred and wild specimen. Moreover, the crayfish in our experiment were kept in similar conditions and were fed the same food so the effects of the lab environment on their microbiome should be quite similar. As described above, we think lab studies have merit even though they do not reflect the outside situation.

• Therefore this piece of work is not scientifically sound especially since P.leniusculus is not included and also because references to work done on plague and crayfish are not correct and seems to have been chosen randomly rather chosen which paper showed it first.

o We disagree with the statement that our work is not scientifically sound. Our study did not include Pacifastacus leniusuculus but that does not mean that the results of our study were not obtained through thorough scientific procedures. As described above we wrote more about P. leniusuculus in our introduction and conclusion. We think that our results should be shared and we hope that our paper will be a starting point for other studies on interspecific agonistic interaction between Astacus astacus and invasive species. We agree that our references could be improved and we did so. 

Reviewer 2: 

• Although potentially interesting the authors make very far-reaching conclusions when considering the limited amount of data those are based on. It is suggested that the text is modified somewhat with respect to the preliminary nature of their respect to reflect this. Also, the discussion regarding changes of pathogen resistance among indigenous crayfish needs some clarifications.

o Response: We agree with reviewer #2 that our conclusions might be too far-reaching and we changed them to so they are better in line with our data. We changed lines 27- of our abstract and lines 273-277 of our discussion, as described in our response to reviewer #1. Furthermore, we changed lines 40-41 of our abstract “The results of this study imply that in case of resistance to the crayfish plague, A. astacus could displace these invasive crayfish from its native habitat.” into “The results showed that A. astacus triumphs over F. limosus and P. acutus in agonistic encounters and in competition for shelter. In turn, P. acutus dominates F. limosus in staged encounters and shelter. In possible future situations were crayfish plague does no longer eradicate noble crayfish populations, our results show that the native noble crayfish might still have a promising future when confronted with invasive species.”. Moreover, we clarified our discussion regarding the resistance of A. astacus against the plague, as described in response to reviewer #1. 

• In total a modest number of 10 Astacus astacus individuals were tested for aggressiveness towards about similar numbers of the other two species. Presumably they were related and of similar age. While it seems plausible that this species is more aggressive than the other two, testing other individuals at other life cycle stages with respect to moulting, egg-bearing, age etc could have resulted in other outcomes and being of relevance for how the species interact with each other. The authors strongly stated conclusion appears a bit to simplistic.

o Response: We agree with the reviewer that testing other life stages is also very relevant for the whole life-cycles. The outcomes of the interactions of other life stages may indeed yield different results but that would not affect the outcomes of the present study. Our crayfish were in intermoult stage. We aimed to focus on the differences between the species and only included one life stage. Future research could include other life stages as well but also interactions of animals that differ in size. The study design used by Kozák et al. (2007), in which juvenile crayfish of two species were reared together, seems to offer a very interesting way to study agonistic interaction at different life stages, including moulting. We added the following sentence to our discussion (lines 245-247) “How these differences in juvenile development affect agonistic interactions between A. astacus and invasive crayfish remains unclear and can be studied in experiments in which juveniles are reared together (33).” Furthermore, as described above in response to reviewer #1 we reformulated our conclusions and made sure they are not too strongly stated anymore.

• The authors are stating that native crayfish are becoming more resistant towards crayfish plague. Available data – as based on the papers cited in the ms and a few more – show the existence of A. astaci strains with a lowered virulence (reference 9) and a few well-documented cases of variations in resistance between different populations of indigenous crayfish populations (especially reference 29). There are so far no experimental demonstrations of a change towards a higher A. astaci resistance among native populations although such changes may well take place in the long run. However, at this stage the conventional epidemiological model that a pathogen with a dramatically shorter generation time than the host is quicker to adapt its virulence (i.e. to lower it and not go extinct) should be over-looked. Some of the papers the authors cite in their discussion in this context, e.g. ref 28, actually argue for that pathogen virulence changes are a major factor at play here. The existence of a melanisation reaction could, as the authors suggest, be interpreted as a sign of host resistance but the other side of the coin is that melanisation can be a consequence of a lower parasite virulence that enables the host to mount a defence.

o Response: We thank the reviewer for his/her clarifications on the discussion on whether it is more likely for A. astacus to become more resistant to the plague or for the plague to become less virulent. References 34, 28 and 35 of our manuscript mention both options, but indeed, seem to favour the lower virulence theory over the increased resistance theory. We rephrased lines 237-240 of our manuscript: “This suggests an enhancement of the immune response of these species to A. astaci, which can enable the melanin encapsulation mechanism that allows North American A. Astaci carrier so live with the pathogen.” into “These findings indicate that perhaps because of lower A. astaci virulence or maybe even because of an enhanced immune response in the crayfish, the melanin encapsulation mechanism that enables North American A. astaci carriers to live with the pathogen, can take place in these populations (34,28,35).” 

Reviewer #3: 

• 1. The objective and hypothesis of the study need to be more clearly written and emphasized in the introduction.

o Response: We rewrote the last paragraph of our introduction to clarify our objective and hypotheses (lines 83-103) “In the laboratory, paired species experiments were performed to compare outcomes of interspecific agonistic encounters (16-18,20-25) and the ability of competing species to obtain shelter (16,19,20,22). We hypothesise that F. limosus will be less successful than the other two species in agonistic interactions and competition for shelter because F. limosus is known to be low in aggression (26,27 Aggressive behaviour is common in both A. astacus (18) and P. acutus (20) but there is no literature available on dominance in agonistic behaviour and competition for shelter between the two species. Therefore, we hypothesize that both species have equal changes in aggressive encounters and shelter occupancy.

• 2. Line 51. It is said a notorious vector of the crayfish plague…. Since all vectors are notorious, I would erase this adjective or search for another one: well-known or well-studied.

o Response: We agree that all vectors of the crayfish plague are notorious and we changed “notorious” into “well-studied”. 

• 3. Line 58 Europe`s most common and mostly highly valued indigenous crayfish is the noble crayfish…. I would said economically most valuable.

o Response: We changed “most highly valued” into “economically most valuable”. 

• 4. There is no section for figure legends and instead there is a figure legend for Fig 1 inserted in the text in pag 5.

o Response: In the PLOS ONE style guidelines we read that “Each figure caption should appear directly after the paragraph in which they are first cited”. Therefore, we added the figure title and legend in the text. We removed the dots from the figure labels and changed the colons after the labels into dots, e.g. changed “Fig. 1:” in “Fig 1.” and reformatted the legends so they are below the title and not in bold anymore. (Lines ) We moved the caption of Fig 3 one paragraph down so it is placed immediately after the paragraph in which it is cited first. Although we could not find the requirement for a section for figure legends in the style guidelines, we added a section for figure legends below our references

• 5. Line 61 to 63. Be aware that the pathogen is not becoming less virulent. This only appears to apply for one haplotype of the pathogeni (haplotype -A but no for others, e.g, B, D1, D2 and E)

o Response: Yes, this is true. Therefore we changed lines xxx, as detailed in our response to reviewer #1. 

• 6. There are only three figures in the article, and there is no need for tables to be in a supplementary materials.

o Response: We agree that there is enough space for tables in the manuscript. However, we do not feel that moving the tables from the supplementary materials to the manuscript would add a lot to the manuscript. The tables in the supplementary materials contain the raw data for the figures and the outcomes of a statistical test. We want to make this data available but feel it is right to have it in the supplementary materials instead of in the main text. 

• 7. The quality of figures is low and there is no section for figure legends

o Response: We improved our figures and added a section for figure legends below our references

• 8. The discussion misses the fact that the two predominant invasive crayfish species are Procambarus clarkii and Pacifastacus leniusculus and that similar studies considering these species are needed. It is true that they are mentioned but I would suggest further discussed this fact.

o Response: We extended our discussion on other, potentially more aggressive, invasive species and stated that future studies should take P. clarkii, P. leniusculus and Cherax destructor into account, as described in our response to reviewer #1.

• 9. Make sure the scientific names of species are in italics in all the text, figures and references.

o Response: We than the reviewer for this suggestion and have italicised the scientific names in figures and references.

• 10. Please write the full names of the species when a sentence starts with a species name.

o Response: We apologise for not having done this and changed “F. limosus” into “Faxonius limosus” in line 52, “A. astacus” into “Astacus astacus” in line 79, line 107 and line 242 and “P. acutus” into “Procambarus acutus” in line 196

• 11. A number of references can be improved. I suggest the following:

• - Statements for Lines 46, a more detailed and specific reference are in reviews by:

o Söderhäll, K., and Cerenius, L. (1999). The crayfish plague fungus: history and recent advances. Freshw. Crayfish 12, 11–35.

o Cerenius, L., Andersson, M.G., and Söderhäll, K. (2009). ”Aphanomyces astaci and crustaceans,” in: Oomycete Genetics and Genomics: Diversity, Interactions and Research Tools, eds. K. Lamour and D Kamoun (New Jersey: John Wiley & Sons Inc.), 425-433.

o Rezinciuc, S., Sandoval-Sierra, J.V., Oidtmann, B., and Diéguez-Uribeondo, J. (2016). “The biology of crayfish plague pathogen Aphanomyces astaci: Current answers to most frequent questions”, in: Freshwater Crayfish – A Global Overview, eds. T. Kawai, Z. Faulkes, and G. Scholtz (London, UK, Taylor and Francis Group, CRC Press), 182–204.

o Response: We would like to thank reviewer #3 for his/her suggestions. We changed our reference (4) into (4) for lines and into (4,8) for lines with (4) “Söderhäll K, Cerenius L. The crayfish plague fungus: history and recent advances. Freshw. Crayfish. 1999;12: 11–35 “ and (8) “Cerenius L, Andersson MG, Söderhäll K. Aphanomyces astaci and crustaceans. In: Lamour K, Kamoun D, editors. Oomycete Genetics and Genomics: Diversity, Interactions and Research Tools. New Jersey: John Wiley & Sons Inc. 2009; 425-433.” We agree that this review article and this book chapter provide more detail about Aphanomyces astaci and its interaction with its crayfish hosts than our original reference. 

• - Statement starting in line 47 and 49. Original citations are:

• Unestam, T. (1969b). Resistance to the crayfish plague in some American, Japanese and European crayfishes. Rep. Inst. Freshw. Res., Drott. 49, 202-209. doi: n/a

• Unestam, T. (1972). On the host range and origin of the crayfish plague fungus. Rep. Inst. Freshw. Res., Drott. 52, 192-198. doi: n/a

• Unestam, T., and Weiss, D.W. (1970). Host parasite relationship between freshwater crayfish and crayfish disease fungus, Aphanomyces astaci. Responses to injection by a susceptible and a resistant species. J. Gen. Microbiol. 60(1), 77-90. doi: 10.1099/00221287-60-1-77

• Or recent publication:

• Martín Torrijos L, María Martínez Ríos, Gloria Casabella Herrero, Susan B.Adams, Colin R. Jackson, Javier Diéguez Uribeondo. 2021.Tracing the origin of the crayfish plague pathogen, Aphanomyces astaci, to the Southeastern United States, Scientific Reports. 11:9332 | https://doi.org/10.1038/s41598-021-88704-8.

o Response: We would like to thank the reviewer for these suggestions and agree that they are more fitting than our reference (5). Unestam (1969) and Unestam (1972) are the original studies indicating that Aphanomyces astaci probably originates from North America and the study by Martín-Torrijos et al. (2021) provides compelling evidence for the North American origin of Aphanomyces astaci and an interesting discussion. Therefore we changed our reference 5 into (5,6,7) with (5) “Unestam T. Resistance to the crayfish plague in some American, Japanese and European crayfishes. Rep. Inst. Freshw. Res., Drottningholm 1969;49:202-9.”, (6) “Unestam, T. On the host range and origin of the crayfish plague fungus. Rep. Inst. Freshw. Res., Drottningholm. 1972;52:192-8.” and (7)“ Martín-Torrijos L, Martínez-Ríos M, Casabella-Herrero G, Adams SB, Jackson CR, Diéguez-Uribeondo J. Tracing the origin of the crayfish plague pathogen, Aphanomyces astaci, to the Southeastern United States. Sci Rep. 2021;11(9332)”

o 

• - Statement line 52: Original citation for presence of A. astaci in orconectes and its role as vector is: Vey, A., Söderhäll, K., and Ajaxon, A. (1983). Susceptibility of the Orconectes limosus Raff. to the crayfish plague, Aphanomyces astaci Schikora. Freshw. Crayfish 5, 284-291. doi. 10.5869/fc.1983.v5.284

o Response: We would like to thank the reviewer for this reference. We changed our reference for lines 57-58 and for lines 58-58 to (9)“Vey A, Söderhäll K, Ajaxon, A. Susceptibility of the Orconectes limosus Raff. to the crayfish plague, Aphanomyces astaci Schikora. Freshw. Crayfish. 1983;5: 284-291”

• lines 203-207. requieres to be rewritten.

o Response: Lines xxx ”For 4 of these 5 pairs more first attacks and wins and less retreats were observed by A. astacus than by P. acutus. Furthermore, the one A. astacus in the experiment that was shorter than its P. acutus opponent still started more fights, won more fights and retreated less often than its competitor” were changed into “with A. astacus starting and winning more fights and retreating less than P. acutus. Interestingly, there was one A. astacus that was shorter than its opponent but still dominated the encounters” 

We thank you for giving us the opportunity to improve our manuscript and we hope that our edits and responses address the points raised by the reviewers in a satisfactory way. 

Yours sincerely,

On behalf of all authors,

Dr ir Ivo Roessink

Senior scientist Environmental Risk Assessment

Wageningen Environmental Research

Team Environmental Risk Assessment

PO box 47, 6700 AA, Wageningen

The Netherlands

Tel. +31 317 481692

---

## [Decision Letter · Decision Letter 1]

20 Sep 2021

PONE-D-21-20033R1Native European crayfish Astacus astacus competitive in staged confrontation with two invasive crayfish speciesPLOS ONE

Dear Dr. Roessink,

Thank you for submitting your manuscript to PLOS ONE. After careful consideration, we feel that it has merit but does not fully meet PLOS ONE’s publication criteria as it currently stands. Therefore, we invite you to submit a revised version of the manuscript that addresses all the points raised during the review process. The reviewers found that there is a need for more experiments in order to get solid data to merit publication.

We look forward to receiving your revised manuscript.

Kind regards,

Irene Söderhäll

Academic Editor

PLOS ONE

Reviewers' comments:

Reviewer's Responses to Questions

**Comments to the Author**

1. If the authors have adequately addressed your comments raised in a previous round of review and you feel that this manuscript is now acceptable for publication, you may indicate that here to bypass the “Comments to the Author” section, enter your conflict of interest statement in the “Confidential to Editor” section, and submit your "Accept" recommendation.

Reviewer #1: (No Response)

Reviewer #2: (No Response)

2. Is the manuscript technically sound, and do the data support the conclusions?

Reviewer #1: No

Reviewer #2: Partly

3. Has the statistical analysis been performed appropriately and rigorously? 

Reviewer #1: Yes

Reviewer #2: N/A

4. Have the authors made all data underlying the findings in their manuscript fully available?

Reviewer #1: Yes

Reviewer #2: Yes

5. Is the manuscript presented in an intelligible fashion and written in standard English?

Reviewer #1: Yes

Reviewer #2: Yes

6. Review Comments to the Author

Reviewer #1: The authors have decided not to include Pacifastacus leniusculus in their study because they argue that this crayfish is less abundant in Netherlands. P.leniusculus is very abundant in several European countries so the value of including this species is very high. Thus this reviewer still considers that at least this aggressive species should be included and as mentioned by reviewer 2 , 10 crayfish is not many and can be sent easily from several countries neighboring the Netherlands.

Reviewer #2: The authors have amended the manuscript according several points raised in the review process. The major one remains though, the scope of the study with respect to number of crayfish individuals, number of species used (several important introduced species are lacking), life cycle stages, genetic background etc remains very limited. It remains doubtful whether that much useful information can be derived from such a restricted study, even if the isolated experiments themselves are properly designed and analysed. In addition, although the text is improved compared to the first version, it still provides the doubtful imaginary that native crayfish are close to a comeback and are overcoming the threat of the crayfish plague. In essence it still doubtful whether publishing at this stage should be encouraged.

7. PLOS authors have the option to publish the peer review history of their article (what does this mean?). If published, this will include your full peer review and any attached files.

Reviewer #1: No

Reviewer #2: No

---

## [Author Response · Author response to Decision Letter 1]

3 Oct 2021

Dear reviewers,

Thank you for your reply and the remarks. We greatly appreciate these efforts to help to improve the document. Reading the comments of both reviewer 2 and previously reviewer 3, we conclude that that the research has been performed adequately and have been statistically analysed accordingly. As a result, we feel that we comply with the mission of PlosOne to publish all methodologically and ethically rigorous research. 

As reviewer 2 considered our writing style, still a bit too strongly formulated at points we have tried to reformulate this in the revised manuscript. We would like to point out that nowhere in the manuscript we claim that agonistic interactions studied in the laboratory are the sole explanation to success in the wild. Surely, more factors are at play here. We simply want to state that if native crayfish can by-pass the huge challenge of the crayfish plague (and some first signals have been published and have been referred to in the manuscript) a whole new interesting set of interdependencies will arise that requires investigation. This is exactly what we did in our manuscript, where we made a first step and investigated agonistic interactions and shelter occupancy between invasive and native crayfish. Based on these two factors, there is reason to be optimistic. However, further research on other factors is definitely required and we certainly are not heralding a comeback of native crayfish populations based on these findings. 

Furthermore, we do not agree with reviewer 1 that the omission of additional experiments with Pacifastacus leniusculus is a reason refrain from publishing the manuscript. Some of our reasons for this are:

1) in a European setting there are quite a number of combinations of pairing invasive and native crayfish species possible. It is not possible to report all of these combinations in one paper and as such any reported combination is valuable and progresses our knowledge (as agreed by reviewer 3 previously). 

2) The competition between A. astacus and P. leniusculus has already been addressed by Söderback (1995). Although this describes replacement in a lake setting, the already described aggressive nature of P. leniusculus makes an additional investigating in a laboratory setting less interesting. Hence a selection of other invasive species is more relevant.

3) Acquiring invasive species from abroad is in The Netherlands illegal under the EU Regulation 1143/2014 (Union list). The administration required to enable this for research purposes involves cooperation on national level from both sending and receiving country. Needless to say that this is not a viable option. Perhaps that the location of reviewer 1 holds an exception to these restrictions, but the suggestion to import 10 individuals to The Netherlands is technically of a criminal nature, unfortunately. We nevertheless appreciate the fact that the amount of 10 individuals is considered acceptable. 

We hope that you agree to our argumentation you’ll accept our revised manuscript for publication as a research article in PLOS ONE. 

On behalf of all authors,

Dr ir Ivo Roessink

---

## [Decision Letter · Decision Letter 2]

16 Nov 2021

PONE-D-21-20033R2Native European crayfish Astacus astacus competitive in staged confrontation with the invasive crayfish Faxonius limosus and Procambarus acutusPLOS ONE

Dear Dr. Roessink,

Thank you for submitting your manuscript to PLOS ONE. After careful consideration, we feel that it has merit but does not fully meet PLOS ONE’s publication criteria as it currently stands. Therefore, we invite you to submit a revised version of the manuscript that addresses the points raised during the review process.

It is necessary then to follow the reviewers suggestions. 

We look forward to receiving your revised manuscript.

Kind regards,

Irene Söderhäll

Academic Editor

PLOS ONE

Journal Requirements:

Reviewers' comments:

Reviewer's Responses to Questions

**Comments to the Author**

1. If the authors have adequately addressed your comments raised in a previous round of review and you feel that this manuscript is now acceptable for publication, you may indicate that here to bypass the “Comments to the Author” section, enter your conflict of interest statement in the “Confidential to Editor” section, and submit your "Accept" recommendation.

Reviewer #1: (No Response)

2. Is the manuscript technically sound, and do the data support the conclusions?

Reviewer #1: Partly

3. Has the statistical analysis been performed appropriately and rigorously? 

Reviewer #1: N/A

4. Have the authors made all data underlying the findings in their manuscript fully available?

Reviewer #1: Yes

5. Is the manuscript presented in an intelligible fashion and written in standard English?

Reviewer #1: Yes

6. Review Comments to the Author

Reviewer #1: The authors argue that it is not possible to import the very well spread P.leniusculus to Netherlands for scientific purposes which this reviewer finds surprising since there must be one way of importing an organism in small numbers for scientific purposes in all EU countries simply by asking for permission from the correct authority. One option could also have been to visit a laboratory in a country with this crayfish (P.leniusculus) and perform these experiments in that country. Therefore this reviewer find little value in this manuscript since one of the most common invasive species P.leniusculus is not included in these studies. As a consequence it is necessary to include a statement in the text that some other invasive species such as for example Pacifastacus leniusculus was not tested for the reason that .......... This reviewer also strongly agrees with the notion from another reviewer that being a bit more aggressive to invasive species will not at all help native species to survive in waters with invasive species since Aphanomyces astaci , the crayfish plague, is present in all invasive species. Thus in this case this manuscript gives more hope than reality shows and will show. Therefore this has to be included in the text as well i.e that the crayfish plague is most likely to be transferred from invasive crayfish to native crayfish regardless of behavior. If these short-comings of this manuscript is not clearly stated and included in the manuscript it should be rejected.

7. PLOS authors have the option to publish the peer review history of their article (what does this mean?). If published, this will include your full peer review and any attached files.

Reviewer #1: No

---

## [Author Response · Author response to Decision Letter 2]

10 Jan 2022

We have taken the comments of reviewer 1 serious and have made the following amendments:

- highlighted that signal crayfish (P. leniusculus) is an important invader in Europe (L74-78).

- explained that P. lenisculus was not part of our study, since its interaction with A. astacus has already been established in previous research and it was as well difficult to acquire at the time of experimentation (L74-78).

- More clearly explained that more factors than aggressive interaction and shelter occupancy determine the success of populations in the field. And that our research does not claim that crayfish plague is no issue of importance but that once this is not a determining factor anymore, native species do not necessarily loose interactions with invaders by default (L277-284).

---

## [Editor Report · Decision Letter 3]

13 Jan 2022

Native European crayfish Astacus astacus competitive in staged confrontation with the invasive crayfish Faxonius limosus and Procambarus acutus

PONE-D-21-20033R3

Dear Dr. Roessink,

We’re pleased to inform you that your manuscript has been judged scientifically suitable for publication and will be formally accepted for publication once it meets all outstanding technical requirements.

Kind regards,

Irene Söderhäll

Academic Editor

PLOS ONE
---

## [Editor Report · Acceptance letter]

17 Jan 2022

PONE-D-21-20033R3 

Native European crayfish *Astacus astacus* competitive in staged confrontation with the invasive crayfish *Faxonius limosus* and *Procambarus acutus*

Dear Dr. Roessink:

I'm pleased to inform you that your manuscript has been deemed suitable for publication in PLOS ONE. Congratulations! Your manuscript is now with our production department. 

Kind regards, 

on behalf of

Dr. Irene Söderhäll 

Academic Editor

PLOS ONE